# Effect of School-Based Food and Nutrition Education Interventions on the Food Consumption of Adolescents: A Systematic Review and Meta-Analysis

**DOI:** 10.3390/ijerph191710522

**Published:** 2022-08-24

**Authors:** Gidyenne Christine Bandeira Silva de Medeiros, Kesley Pablo Morais de Azevedo, Daniel Garcia, Victor Hugo Oliveira Segundo, Ádala Nayana de Sousa Mata, Anny Karoliny Pinheiro Fernandes, Raquel Praxedes dos Santos, Débora Danielly Barros de Brito Trindade, Isabel Morales Moreno, Daniel Guillén Martínez, Grasiela Piuvezam

**Affiliations:** 1Systematic Review and Meta-Analysis Laboratory (Lab-Sys), CNPq-UFRN, Department of Nutrition, Federal University of Rio Grande do Norte (UFRN), Natal 59078-970, Brazil; 2Postgraduate Program in Public Health, Federal University of Rio Grande do Norte (UFRN), Natal 59078-970, Brazil; 3Department of Physiotherapy, UCAM Universidad Católica de Murcia, 30107 Murcia, Spain; 4Multicampi School of Medical Sciences of RN, Federal University of Rio Grande do Norte (UFRN), Natal 59078-970, Brazil; 5Department of Nutrition, Federal University of Rio Grande do Norte (UFRN), Natal 59078-970, Brazil; 6Department of Nursing, UCAM Universidad Católica de Murcia, 30107 Murcia, Spain; 7Systematic Review and Meta-Analysis Laboratory (Lab-Sys), CNPq-UFRN, Postgraduate Program in Public Health, Federal University of Rio Grande do Norte (UFRN), Natal 59078-970, Brazil

**Keywords:** adolescent, education, nutrition, school, randomized clinical

## Abstract

The school is a favorable environment for the development of interventions to prevent obesity. The objective of this systematic review is to evaluate the effects of school-based food and nutrition education interventions on adolescent food consumption. The literature search was conducted on databases: MEDLINE/PubMed, Embase, Scopus, ERIC, Science Direct, Web of Science, Cochrane, LILACS, and ADOLEC. The following research strategies were focused on: population (adolescents), intervention (food and nutrition education), outcome (food consumption), and study design (clinical trial). The Preferred Reporting Items for Systematic reviews and Meta-Analyses (PRISMA) statement guidelines were followed and all stages of this review were performed by two researchers and, when necessary, a third researcher resolved discrepancies. Included studies are randomized clinical trials (RCT). A total of 24 articles were included for review and 11 articles in meta-analysis. In the evaluation of the general effects, there was a significant effect (mean difference (MD) for fruit consumption (MD = 0.09, CI 0.05, 0.14) in serving/day; and for vegetables (MD = 0.59, IC 0.15, 1.03) at times/week. In the consumption of FV (fruits and vegetables), there was no significant effect (standardized mean difference (SMD) of interventions in their consumption (SMD = 0.00, 95% C1 −0.11, 0.11). The evidence available in this review and meta-analysis concludes that food and nutrition education interventions in schools presented favorable results in the food consumption of adolescents. Registered on the PROSPERO database (CRD42019116520).

## 1. Introduction

The prevalence of obesity has increased worldwide, not only among adults but also among children and adolescents as well. In 2016, 18% of children and adolescents aged 5–19 in the world were overweight or obese [1,2]. These data are important because obesity in adolescence is a condition that presents itself as a risk factor for becoming an obese adult [3], in addition to being associated with the development of chronic diseases, leading to a marked increase in morbidity and mortality from cardiometabolic disease in adulthood [4]. The treatment of obesity is complex, and many intervention trials are ineffective in the treatment of childhood and adolescent obesity [5]. These facts reinforce the importance of developing intervention strategies to prevent overweight and obesity in adolescence.

Among the possible strategies are food and nutrition education interventions in schoolchildren. The school is a favorable environment for the development of this type of intervention, with the objective of preventing overweight and obesity among adolescents, as it is in this space that they spend most of their time and are involved in a process of learning and changing of behaviors [2,6,7].

A systematic review identified that interventions in nutrition education, carried out with children aged 2 to 19 years, who were more likely to succeed were those that had a multicomponent approach, appropriate to age and adequate duration (≥6 months), which involved family and that ensured fidelity and proper alignment between stated objectives [8].

A multicomponent approach has more than one active component associated with the main component of the intervention. In school-based food and nutrition education interventions, for example, the intervention inserted in the school curriculum can be the main component and, associated with it, actions are carried out in other components (environment, family, teacher training, etc.), to have a more holistic scope of intervention.

There are a small number of systematic review studies that assess interventions specifically for adolescents, such as the study of Meiklejohn, Ryan, and Palermo (2016) on multicomponent interventions that encompassed food and nutrition education. In this study, however, very specific criteria were identified, such as the specification of the intervention model, publication time, and the inclusion of randomized studies carried out in developed countries, which may limit knowledge about the impact of different strategies carried out around the world [9].

It is important that food and nutrition education interventions designed specifically for adolescents, in different social and economic contexts, be evaluated to determine the strategies that have shown favorable results for healthy eating. Expanding this knowledge allows the identification of intervention models that can be improved and replicated with other adolescents. Therefore, the objective of this systematic review is to evaluate the effects of school-based food and nutrition education interventions on adolescent food consumption.

The results presented in this systematic review and meta-analysis study showed several experiences aimed at the adolescent school public on the theme of food and nutrition education and showed positive results regarding the changing of habits and choices for healthy foods. From this perspective, the findings have the potential to influence new investigations with important recommendations for intervention models in food and nutrition education aimed at researchers and governmental public policies in the areas of education and health to collaborate with the collective health of school-aged adolescents through more effective and lasting interventions

## 2. Materials and Methods

This systematic review was conducted following the Preferred Reporting Items for Systematic reviews and Meta-Analyses (PRISMA) statement guidelines [10] and details of the methodology were published previously [11].

### 2.1. Search Strategy

The literature search was conducted in June 2019 on databases: MEDLINE (via PubMed), Embase (via OVID), Scopus (via Elsevier), ERIC—Education Resources Information Center, Science Direct (via Elsevier), Web of Science—Main Collection (Clarivate Analytics), Cochrane Central Register of Controlled Studies (CENTRAL), LILACS (via Biblioteca Virtual em Saúde—BVS), and ADOLEC (via Biblioteca Virtual em Saúde—BVS). There was no time and language limitation.

The searches focused on the following four key elements: population (children, adolescents), intervention (food and nutrition education interventions school-based), outcome (food consumption), and study design (clinical trial). The search strategy depicting the combination of keywords used for the literature search was published before [11].

### 2.2. Eligibility Criteria

Included studies are original articles reporting randomized clinical trials (RCT). These trials report the effect of the school-based food and nutrition education interventions on adolescent food consumption. Studies that reported only the percentage of students who consumed a certain food or not were included. Additionally, studies that reported only the consumption of beverages were not included.

It was not necessary to insert other study designs (non-randomized clinical trials, or controlled before-after studies), since the RCTs retrieved in the searches were sufficient to answer the research question.

The included studies involve adolescent intervention participants, considering the World Health Organization definition, people aged 10–19 years [12]. The studies the participants were adolescents with physical disabilities, intellectual disabilities, endocrine disorders, chronic diseases (cardiovascular diseases, diabetes, and obesity), and pregnant were excluded. Moreover, gray literature, review articles, and articles that did not include clear information about intervention methodology assessed children and adolescents as a single group and evaluated only nutrients and not food were also excluded.

### 2.3. Data Extration

The articles selected in the research bases were inserted in the Rayyan Web application [13]. Then, using the application, at least two independent authors reviewed the titles and abstracts. The articles that met the inclusion criteria were requested for review by full reading by two independent researchers. Any disagreements between the two authors were resolved by discussion with a third reviewer to reach a consensus.

Data extraction was performed by two authors. Extracted data included study design, participants’ characteristics, control group, intervention characteristics, dietary assessment, outcome (food consumption), and analysis methods. Studies that have published an intervention protocol article, when necessary, were also consulted in data extraction. Any disagreements in the extracted data between the two authors were resolved by discussion and re-examination of the article to reach a consensus.

### 2.4. Risk of Bias Assessment

The methodological quality of the studies was assessed by the Cochrane risk of bias tool for randomized trials (RoB 2) [14]. This is the most used tool to assess the risk of bias in randomized studies. The RoB 2 has an evaluation in five different domains (randomization process, deviations from intended interventions, missing outcome data, measurement of the outcome, and selection of the reported result). The answers lead to judgments of “low risk of bias”, “some concerns”, or “high risk of bias” [14]. In the end, the judgments in each domain lead to a general judgment of risk of bias for the result being evaluated, which in this review is food consumption.

Two independent researchers carried out the evaluation and, when necessary, a third researcher was consulted. Thus, the articles were classified as low, high, or some concerns, using predetermined criteria.

### 2.5. Summary and Data Synthesis

The extracted data were presented in a narrative approach to summarize the results of this review. The results of the intervention effect on food consumption were related to baseline values and the first post-intervention assessment. The significance of the differences in the outcome measures reported in the included studies was indicated using *p* values.

Data synthesis of food consumption of fruits, vegetables, and FV (fruits and vegetables) was performed. The results of consumption before and after the interventions were extracted to calculate the delta of variation (Δ) and the standard deviations of variation for the intervention group and the control group.

To standardize the consumption measures, the most frequent measure presented by the studies was used. Thus, to evaluate the consumption of FV, we included the studies that presented the results of consumption in serving/day or score.

The data were in grams and were converted into serving, considering an average serving of 100 g. To evaluate fruit consumption, studies that evaluated consumption in serving/day were included. The data were presented in serving/week and were converted into serving/day.

The evaluation of the heterogeneity between the studies was verified by the standard X2 test with a significance level of 0.05. The I2 statistic was calculated, wherein a value of 0% indicates no observed heterogeneity, while the values above 50% indicate a substantial level of heterogeneity.

To calculate the total effect size of the included studies, the random-effects model was used. Meta-analysis of included studies was performed using Review Manager 5.3 (The Nordic Cochrane Centre: Copenhagen, Denmark).

## 3. Results

### 3.1. Literature Search Results

In the search for articles in the databases, 5010 studies were obtained. In addition, in a manual search, the references of the systematic studies retrieved in the searches were examined. Thus, 14 articles were included for evaluation of title and abstract. After excluding the 738 duplicate studies, the remaining 4286 titles and abstracts were read and evaluated by the researchers. In addition, systematic studies were examined, and data found were included in the review process at this stage. After reviewing the title and abstract, 342 studies were selected for a full reading. A total of 24 articles were included for review [15,16,17,18,19,20,21,22,23,24,25,26,27,28,29,30,31,32,33,34,35,36,37,38] and 11 articles in meta-analysis [17,19,23,24,26,27,30,32,33,34,38]. A PRISMA diagram depicting the flow of literature search and article selection is presented in Figure 1.

### 3.2. Methodological Quality of Included Studies

In the evaluation of the overall bias of ROB 2, seven articles were evaluated as ‘low’, three as ‘high’ risk of bias, and 14 as can express ‘some concerns’ for the outcome of food consumption (Table 1). The summary of the risk of bias by the Cochrane risk of bias tool for randomized studies (RoB 2) is shown in Figure 2.

The “randomization process” and “selection of the reported result” domains were the ones with the highest number of studies with “some concern” risk of bias. In the “randomization process” domain, half of the studies did not have a detailed description of the processes of generating and hiding the random sequences. In “selection of the reported result”, in 54% of the studies, the authors did not report or did not provide sufficient detail to classify the risk of bias regarding the data that produced the food consumption outcome. There is no information to indicate that these data were analyzed according to a pre-specified analysis plan.

The results of each domain of the RoB 2 tool, per article included, are in the Appendix A.

### 3.3. Study Characteristics

The RCTs that were included in this review are publications from 1997 to 2019 and from 14 different countries. From the USA, there were nine studies; from Iran, Greece, and the Netherlands, there were two studies from each country; and from other countries (Italy, Norway, Brazil, Trinidad Tobago, UK, Belgium, Finland, Ecuador and China), there was a study each.

A summary description of the school-based food and nutrition education intervention studies is presented in Table 1.

### 3.4. Intervention Characteristics

The interventions were based on different theories and models. The most frequent were: Social Cognitive Theory (SCT) [18,20,22,31,32,33,38] and Theory of Planned Behavior (TPB) [17,19,27,29]. Ten studies did not claim to have been based on any theory or model [15,16,21,23,24,25,26,30,35,36].

Most studies (67%) performed the intervention with more than one component [18,19,20,21,22,23,24,25,26,28,30,32,33,34,35,38]. In nine studies, the intervention had the “school environment” component, with actions such as change in school meals, educational posters by the school, organization of events, and offering fruits and vegetables [20,21,22,25,28,32,33,35,38]. The “family” component was present in 13 studies, with actions such as sending messages and leaflets to parents, events at school with the participation of the family and offering fruits and vegetables [20,21,22,23,24,25,26,28,30,32,33,34,38]. Another component presented was the “Teacher Professional Development”, four studies presented their actions, a specific component for training and support for the teacher to develop the intervention [18,19,20,31]. Finally, in another eight studies, the intervention was centered only on the student, without intervening through other components [15,16,17,27,29,35,36,37]. In four studies, the practice of physical activity was combined with intervention in nutritional education [21,26,35,37]. The details of the studies included in the systematic review are in the Appendix A.

### 3.5. Food Consumption

Food frequency questionnaires (QFF) and 24 h recall (R24 h) were used to assess food consumption. The consumption of fruits and vegetables was assessed by most studies (75%), in aggregate form FV (fruits and vegetables) [17,25,26,32,35,36,38], fruits [18,19,20,22,23,24,27,29,30,33,34,35] and/or vegetables [16,18,20,22,24,29,30,33,35]. Another three studies presented the evaluation of the consumption of a larger group of foods in which fruits and vegetables were also inserted, called healthful eating (score) [31], healthy food [21], semiquantitative food frequency scores [15], and consumption score of fruit, vegetables, dairy products, breakfast, dessert, fried food and soft drinks [28].

For FV consumption, two studies showed significant post-intervention differences [17,32]. In the study by Gratton, Povey, and Clark-Carter (2007) [17], two forms of intervention were tested: the volitional intervention in which children were encouraged to write down how, where and when they would eat five portions of fruit and vegetables for the next 7 days; and the motivational intervention, in which children received ‘health education activity sheet’, seeking to stimulate the increase in participants ’consumption of fruit and vegetables to five portions a day. The control group received a volitional intervention focused on school homework rather than fruit and vegetable consumption. The children of the two intervention groups showed a significant increase (*p* < 0.001) in the consumption of FV, although only the volitional intervention demonstrated a significant increase over the control intervention.

In assessing the overall ROB 2 bias for FV consumption, this study by Gratton, Povey, and Clark-Carter (2007) [17] was rated as having a “low” risk of bias in the domains, “deviations from intended interventions”, “missing outcome data”, and “measurement of the outcome”. The domain “randomization process” was evaluated as having “some concerns” because it did not describe how the randomization process took place, and the domain “selection of the reported result” having as “some concerns”, as the authors do not provide sufficient details to classify the risk of bias, leading being classified in the overall bias as having “some concerns”.

The second study was the Teens Eating for Energy and Nutrition at School (TEENS) study. It showed a significant difference (*p* < 0.05) in the group of interventions “peer leaders plus classroom curriculum plus school environment interventions” [32].

Among the methodologies used in the TEENS study were small-group activities and discussions; audiotape; low-fat convenience snacks distributed for tasting during every session; hands-on fruit and vegetable snack preparations and tasting. “Parent Packs” were sent to parents or guardians, which contained activities and intervention-related messages. The school environment intervention included taste testing of fruits, vegetables, and lower-fat foods, increased availability of appealing fruits and vegetables on the lunch line, and increased availability of good-tasting lower-fat snacks on the a la carte line. A la carte lines and vending machines placed posters comparing fat and sugar in the snack choices. Table tents and posters promoting fruits, vegetables, and lower-fat foods were also exhibited as prize raffles for students taking fruits and vegetables on the lunch line [32].

This study was assessed as having a “low” risk of bias in all domains, except the domain “selection of reported result”, in which it was assessed as having “some concerns” because the authors do not provide sufficient detail to classify the risk of bias, leading to being classified in the general domain bias as having “some concerns”. Among the 11 studies that evaluated fruit consumption [20,21,23,24,25,29,30,32,34,35,36], five studies showed a significant difference after the intervention. They are: the Krachtvoer program [23,27], the HEIA (Health In Adolescent) Study [33], PAPPAS (*Pais, Alunos e Professores para uma Alimentação Saudável*) [34], and the VYRONAS (Vyronas Youth Regarding Obesity, Nutrition and Attitudinal Styles) Study [24].

Two versions of the Krachtvoer program were included, and in both, there was a significant difference in fruit consumption. The first version (2008), regarding fruit consumption in servings per day, presented, adjusted for age, sex and baseline, the β value for the difference between the intervention and control groups at T1 = 0.043, *p* < 0.05 two tailed [23]. In the second version (2012), favorable short-term and long-term intervention effects were found for fruit frequency and yesterday’s fruit consumption (servings per day). The value of the short-term effect was β = 0.048 (0.023–0.053), (*p* < 0.001) and the long-term effect was β = 0.033 (0.017–0.048), (*p* < 0.001) [27]. Unlike food and nutrition education interventions that seek to bring knowledge with the aim of changing behavior towards healthy eating, the Krachtvoer program sought to achieve behavior change based on principles of behavior change theories [23,27].

As for vegetable consumption, when evaluated separately from fruit consumption, only one study showed a significant increase in consumption among adolescents (*p* < 0.01) [16]. This study used a board game called Kalèdo. There were 24 weeks in which adolescents were subjected to game sessions of 15 to 30 min, once a week. The intervention aimed to generate changes in nutritional knowledge and eating behavior [16].

Of the studies in which the consumption of fruits and vegetables was evaluated in a more comprehensive set of foods, it was found that there was a significant improvement after intervention in the consumption of recommended foods [15,28,31], except for one study that did not present a significant difference—however, it showed a significant reduction in consumption of unhealthy foods (*p* < 0.03) [21].

Other foods that evaluated the effect of the intervention on their consumption were: beans and cookies [34], pork, sausages, salami, yellow cheese, butter, olive oil, honey, jam, cake, and cocoa [30], ready-to-eat breakfast cereals, red meat, poultry and fish [24].

Of these foods, after the intervention, there was a significant reduction in the frequency of daily consumption of cookies (*p* < 0.001) [34]; a significant increase in poultry consumption [24], and a significant decrease in red meat [24] consumption in the intervention.

In the study that evaluated the consumption of pork, sausages, salami, yellow cheese, butter, olive oil, honey, jam, cake, and cocoa [30], the boys in the intervention group decreased their consumption of pork (*p* < 0.05), sausage (*p* < 0.05), salami (*p* < 0.01) and jam (*p* < 0.01) after the intervention. Additionally, the girls in the intervention group reduced their consumption of cocoa (*p* < 0.05) and sausages (*p* < 0.05) [30].

### 3.6. Meta-Analysis of Fruit and Vegetables Consumption

In the assessment of food consumption in the inserted studies in this systematic review, a variety of results (consumption of different foods) was verified, characterizing the heterogeneity of outcomes. As the consumption of fruits and vegetables is a marker of healthy food consumption [39] and was present in 75% of the studies, a synthesis of the data for the consumption of these foods was performed.

Four studies [17,26,32,38] were included in the analysis of FV consumption (Figure 3), with seven analysis groups, as the study by Birnbaum et al. (2012) presented three intervention groups: (1) school environment interventions only, (2) classroom curriculum plus school environment interventions, and (3) peer leaders plus classroom curriculum plus school environment interventions; and the study by Gratton et al. (2007) presented two intervention groups: group A with a voluntary intervention, in which participants were asked to form an implementation intention, writing how, where and when they would eat five servings of fruits and vegetables in the next 7 days; and group B, with a motivational intervention that aimed to increase the participants’ consumption of fruits and vegetables to five servings a day, through a change in the children’s beliefs, using two of the steps suggested by Sutton (2002). In the first step, a statement was created to raise awareness of the health benefits of fruits and vegetables, and to create new salient beliefs. Then, in the second step, participants were asked to write down four of their benefits of eating five servings of fruits and vegetables a day [17].

Evaluating the general effects of the interventions included in the meta-analysis on the consumption of FV, there was no significant effect (standardized mean difference (SMD) of interventions in their consumption (SMD = 0.00, 95% C1 −0.11, 0.11).

Regarding fruit consumption (Figure 4) five studies were inserted with six analysis groups since the study by Harens et al. (2007) presented two intervention groups, a group of intervention with parental support and another group with intervention alone. In the evaluation of the general effects, there was a neutral effect (mean difference (MD) for fruit consumption (MD = 0.09, CI 0.05, 0.14) in serving/day.

On the consumption of vegetables, two studies and three analysis groups were inserted (Figure 5), since the study by Hassapidou et al. (1997) presented the results in two groups, with the same intervention, but separated by sex. In this analysis, it was possible to identify a positive and significant effect on the consumption of vegetables (MD = 0.59; 95% CI = 0.15–1.03) for the intervention group (times/week).

Among the analyses performed, there was a substantial level of heterogeneity (I2 = 66%; *p* = 0.007) only for the analysis of FV consumption.

**Figure 3 ijerph-19-10522-f003:**
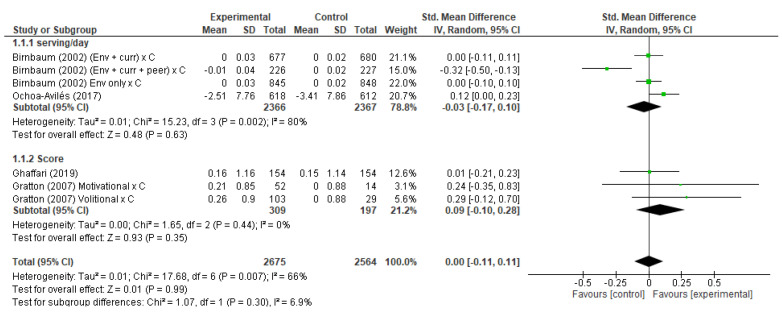
Forest plot of the fruit and vegetable consumption changes [17,32,38].

**Figure 4 ijerph-19-10522-f004:**
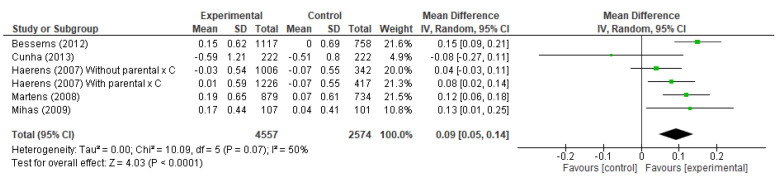
Forest plot of the fruit consumption changes [19,23,24,27,34].

**Figure 5 ijerph-19-10522-f005:**
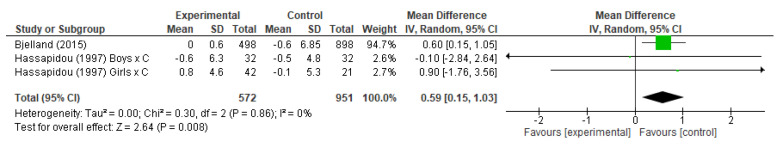
Forest plot of the vegetable consumption changes [30,33].

## 4. Discussion

This review aims to assess the effects of school-based food and nutrition education interventions on adolescent food consumption. In general, these interventions have sought to increase the consumption of healthy foods, especially the consumption of fruits and vegetables, which are markers of a healthy diet [39].

Of the 24 studies included, in 16 the intervention managed to improve the consumption of at least one food or group of foods recommended as healthy food. Of the interventions that assessed both fruit and vegetable consumption separately (n = 8), three showed a significant increase in post-intervention fruit consumption, while there were no significant changes in vegetable consumption [20,24,33]. The results of the meta-analyses were favorable to an increase in the consumption of fruits and vegetables, with a greater effect on the consumption of vegetables. There was a significant effect on the frequency (times/week) of vegetable consumption and the number of servings per day of fruit consumption.

Given these findings, it should be pointed out that children and adolescents tend to have a significantly higher preference for sweetness compared to young adults [40]. In the study by Mennella and Bobowski (2015), it is reported that there is an innate taste preference for sweets and bitter rejection in humans, a consequence of evolutionary selection, favoring the consumption of fruits and tend to have an aversion to the bitter taste present in vegetables [41].

Thus, even if adolescents prefer sweet flavors and often look suspiciously at new foods, they are predisposed to learn from experience, so environmental factors influence food preferences [42]. Considering this evidence, it was observed part of the studies point out that the strategies aimed at tasting healthy foods [19,21,22,23,25,27,31,32,33] showed significant differences in favor of healthy food consumption after the intervention period.

Most of the school-based food and nutrition education interventions included in the present review showed a predominance of multicomponent approaches. According to Yuksel et al. (2020), multicomponent interventions should focus on content, teacher training, and curriculum design to contribute to the development of students’ knowledge, skills, and attitudes [43]. Thus, of the 16 studies that addressed multicomponent interventions [26,28,30,32,33,34,35,38], the components most frequently associated with them was the family component, followed by the teacher component., Furthermore, favorable results were reported for food consumption in the groups submitted to multicomponent interventions, corroborating the hypothesis of the present study [18,21,23,24,28,30,31,32,33,34,38].

The younger the child, the more important factors related to parents and the home. At the adolescent stage, these factors can range from the availability of food in local stores to the extent of advertising to which they are exposed [42]. Although adolescents have greater independence in terms of food consumption, the family remains an important but decreasing source of influence as adulthood approaches [44].

Regarding the teacher component, in general, school-based interventions require the active participation of teachers to ensure the effectiveness of the process. However, the results of this review indicate that 38% of the interventions mentioned training for teachers or provided support material for developing the intervention and encouraging students to change eating habits [18,19,20,21,26,28,31,34,38]. Specifically in one study included in this review, which evaluated the effect of the intervention considering the level of “Teacher Implementation” and “Student Reception”, it was evidenced that when teachers receive support and training throughout the intervention period (e.g., troubleshooting any barriers to implement the curriculum), students become more involved, which makes it more likely to improve students’ psychosocial and behavioral outcomes, and therefore the intervention can be considered more effective because it is better implemented [18].

Furthermore, it is noteworthy that the systematic review study that assessed the effect of multicomponent interventions in developed countries corroborates findings of the present review by concluding that these interventions can have significant impacts on adolescent nutrition when they are theory-based, facilitated by school staff in conjunction with parents and families, and includes changes in the school’s eating environment [9].

In the included studies, we see that more and more researchers have based interventions on different theoretical models. In this context, it is important to plan interventions not only considering the issue of knowledge about food and nutrition but also considering theories and models that can influence food choices. Focusing on the change process, some researchers have used the trans-theoretical model in interventions aimed at changing their lifestyles. A study that sought to systematically review the effectiveness of the trans-theoretical model in multibehavioral interventions to change eating habits and levels of physical activity concluded that the model is a promising strategy for the promotion of healthy lifestyles [45].

New research, considering the stage of behavior change in which adolescents are and seeking to help break down barriers that arise in the middle of the process, could be tested. Of the intervention studies included in this review, only one of them used the trans-theoretical model [29]. In this study, the intake of whole grain bread, whole grains, fruits, and vegetables was evaluated. At the end of the intervention, there was a significant increase in the consumption of whole grain bread, but the magnitude was small. For other foods, there were no significant effects of the intervention [29].

As for the duration of interventions and the effect on food consumption, 93% of interventions of a shorter duration (<1 year) showed significant results, while 60% of interventions of a longer duration (≥1 year) had no significant effects. This fact indicates that the effect of this type of intervention is short lived, requiring continuous work of food and nutrition education to maintain the behavior.

This review and meta-analyses provide us with more specific data on the effect of school-based food and nutrition education interventions aimed exclusively at adolescents on food consumption. The results point us to a path (profile of intervention) that has shown positive effects in changing behavior towards healthy eating. However, it does not define an ideal intervention model to obtain positive effects on adolescent food consumption, but indicates multicomponent interventions are more frequent, and include changes in the environment and the participation of parents, teachers, and school staff, which are important to change the behavior of schoolchildren.

Another important aspect is that the intervention should use a theoretical model about behavioral changes, in addition to knowledge about food and nutrition. Due to the high methodological rigor adopted from the conception of the protocol to the application of the methods and systematization of the results, it is believed that the present study has the potential to guide recommendations focused on the planning of epidemiological studies and public health policies aimed at adolescents and the school environment.

The main strengths highlighted by our findings in this study include the relevance of carrying out multicomponent interventions with strong involvement of teachers and families and with important changes in the school environment aimed at effectiveness in terms of changes in adolescents’ habits regarding food choices for healthy choices and ensuring deep learning on the part of this public. Moreover, the election of a theoretical model for the development of the intervention adapted to the local reality (country, region, and school) also showed to be a path toward the effectiveness of interventions.

Additionally, considering that the present study is a systematic and transparent review, it is appropriate to recognize that the method presents as a limitation a possible omission of studies related to the theme. In this sense, to minimize this bias, all research stages were carried out by two researchers independently, and searches were carried out in nine different databases, in addition to manual searches in the references of the selected studies and systematic reviews of closely related topics. A limitation of the inserted intervention studies is that food consumption was estimated using a food frequency questionnaire or 24 h dietary recall, thus increasing the potential for measurement error and selective underestimation or overestimation of intake, which may compromise the validity of the questionnaire.

Finally, this systematic review and meta-analysis study recommends to researchers and to governmental public policies in education and health that the proposed school-based interventions be carried out based on a theory adapted to the local reality, and that they are multicomponent involving teachers and assistants, the family, and environmental changes that favor healthy food choices by this adolescent school population.

## 5. Conclusions

Food and nutrition education interventions in schools presented favorable results in the food consumption of adolescents. From the quantitative synthesis (meta-analysis), the results were favorable for fruit and vegetable consumption increase but with a greater effect on vegetable consumption. Evidence on fruit and vegetable consumption may be weaker due to the scarcity of comparable studies.

However, we identified a possible problem in maintaining the changes achieved in interventions of a longer duration (≥1 year). This indicates the need for new interventions to also seek to use strategies that pay attention to the factors that contribute to adherence and sustainability of changes in behavior regarding healthy food consumption.

Schools must also adopt the intervention proposals as a school program that accompanies the student through all academic years, contributing to the changes in food consumption become eating habits for adulthood.

## Figures and Tables

**Figure 1 ijerph-19-10522-f001:**
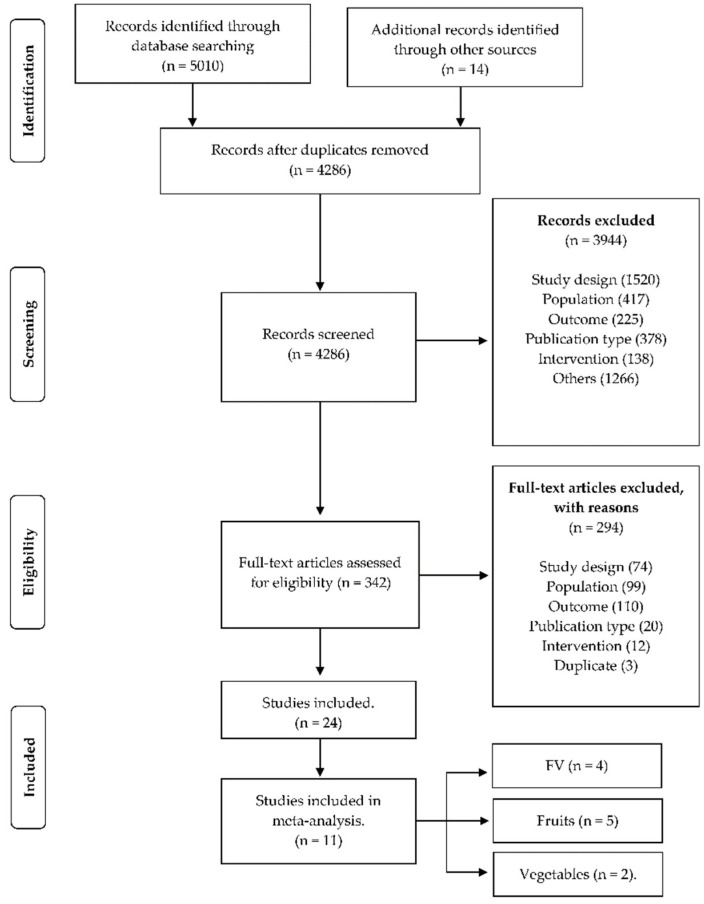
PRISMA flowchart of study selection.

**Figure 2 ijerph-19-10522-f002:**
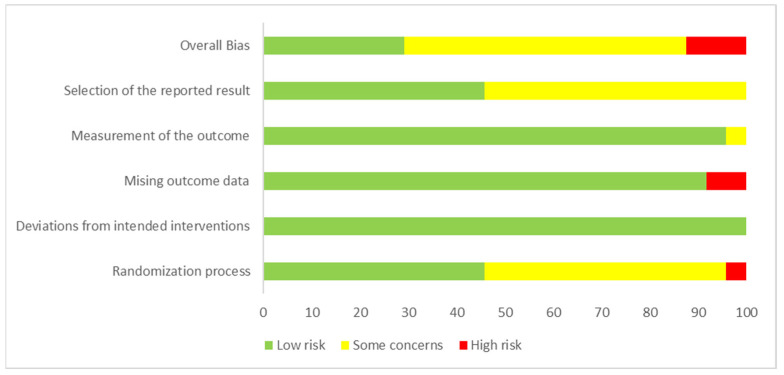
Summary of the risk of bias by the Cochrane risk of bias tool for randomized trials (RoB 2).

**Table 1 ijerph-19-10522-t001:** Summary of school-based food and nutrition education intervention studies.

First Author, Year	Theory or Model	Component	Duration	Foods *	Instrument	Effect	Risk of Bias
Amani, 2006 [15]	___	Classroom	2 months	Semiquantitative food frequency scores (five main food groups of the US Food Guide Pyramid)	FFQ	The food frequency scores were elevated in the education group (*p* < 0.05), but the control group had a non-significant fall in its scores after the campaign (*p* < 0.1)	Some Concerns
Amaro, 2006 [16]	___	Classroom	24 weeks, 15–30-min-long play sessions once a week	Vegetables (serving/day)	FFQ	A significant difference between the treated group and control group at post-assessment (*p* < 0.01)	Some Concerns
Bessems, 2012 [27]	Behavior change theories; Self-Regulation Theory; TPB, Attitude-Social Influence-Self Efficacy Model, and the action planning literature	Classroom	8 weeks	Fruit (serving/day)	FFQ	The significant mean difference between the experimental and control group 0.15 servings at both posttests (*p* < 0.001)	Low
Birnbaum, 2002 [32]	SCT	Classroom curriculum, school environment, and peer leaders. Parent Packs	1 year period, 10 curriculum sessions	FV (serving/day)	FFQ	A significant difference in the group of interventions “peer leaders plus classroom curriculum plus school environment interventions” (*p* < 0.05)	Some Concerns
Bjelland, 2015 ** [33]	Social-ecological framework incorporating elements from SCT	Class, home/parents, school-wide, and leisure time activities	20 months	Fruits and vegetables (times/week)	FFQ	Significant difference between groups post-intervention for fruits (*p* < 0.001), not to vegetables (*p* = 0.46)	Some Concerns
Bukhari, 2011 [31]	SCT and the social-ecological model	Classroom and Teacher Development	19 weeks	Healthful eating (score)	FFQ	There was an overall increase in score of 4.9 points, (*p* < 0.01). Improved scores correlated with reporting increases in eating vegetables as snacks (r = 0.64, *p* < 0.001), preparing healthful snacks for self (r = 0.48, *p* < 0.01), and having sit-down meals with family (r = 0.55, *p* < 0.004)	Some Concerns
Cunha, 2013 [34]	Pedagogy of the Oppressed, by Paulo Freire	Classroom curriculum and parents (participation of the family)	9 months, monthly 1 h sessions in the classrooms	Fruits, beans, cookies, sodas, and juice (per day)	FFQ and 24 h dietary recall	Significant reduction in the frequency of daily consumption of cookies (*p* < 0.001) and sodas (*p* = 0.02) and an increased frequency of consumption of fruits (*p* = 0.04) in the intervention group, compared with that of the control group	Low
Dzewaltowski, 2009 ** [35]	___	Project level, school level, and place level	3 year period	FV, fruit, and vegetables (servings/day)	FFQ	The intervention and control schools did not change differently over time on FV, fruits, or vegetables	Some Concerns
Forneris, 2010 [36]	___	Classroom	12 weeks	FV (score)	FFQ	No significant change patterns were found at follow-up for fruit and vegetable intake	Low
Francis, 2010 ** [37]	Bloom’s mastery of learning model	Classroom	8 months (10 min/day)	Fried food (servings/day) and HFSS (kJ/day)	FFQ	Average reported daily servings of fried foods were significantly lower in the intervention group than in the control group. In multivariate regression equations controlling for age, gender, BMI, and baseline value, the intervention was associated with lower intake levels of fried foods, HFSS, and sodas (*p* < 0.05)	Low
Ghaffari, 2019 [38]	SCT	Student, family, and school levels	1 year period	FV (score)	FFQ	The difference was significant between the intervention and control groups for 2 months after the intervention (*p* < 0.002). No significant difference between the groups before the intervention	Some Concerns
Gratton, 2007 [17]	TPB	Classroom	3 weeks	FV (score)	7-day food diary	Both interventions (volitional and motivational) were found to increase fruit and vegetable consumption significantly (*p* <0.001), although only the volitional intervention demonstrated a significant increase in fruit and vegetable consumption over the control intervention	Some Concerns
Gray, 2015 [18]	SCT and self-determination theory	Student/classroom and Teacher Professional Development	8–10 weeks, 24 lessons were taught	Fruits, vegetables, packaged snacks, fast food, and sweetened beverages	FFQ	Students from the high ‘Teacher Implementation’ classes significantly consumed fewer packaged snacks (*p* < 0.016) and fast food value/combo meals (*p* < 0.047), and smaller sizes of fast food (*p* < 0.001), compared with control students. There was no significant difference in any eating behavior outcomes between medium ‘Teacher Implementation’ classes and the control group	High
Haerens, 2007 [19]	TPB	Student/classroom and teachers	1 year	Fruits (servings/day)	FFQ	The intervention was not effective in increasing self-reported fruit intake	High
Hassapidou, 1997 [30]	___	Student/classroom and parents (leaflets)	10 weeks	Pork, sausages, salami, yellow cheese, butter, olive oil, raw vegetables (salads), apples and pears, citrus fruit, bananas, grapes, kiwi fruit, fruit juice (natural), honey, jam, cake, and cocoa	24 h dietary recall	The intervention did not result in significant changes in the fruits and vegetable intake. The boys in the intervention group decreased their intake of pork (*p* < 0.05), sausages (*p* < 0.05), salami (*p* < 0.01) and jam (*p* < 0.01) after the intervention. Girls in the intervention group reduced their consumption of cocoa (*p* < 0.05), sausages (*p* < 0.05) and animal butter (*p* < 0.01)	High
Hoppu, 2010 [20]	SCT	Food environment and nutritional education (pupils, parents, and teachers)	1 year	Fruits and vegetables (servings)	FFQ	Energy-adjusted consumption of fruit, including berries (g/MJ) remained constant in IG, whereas it decreased in CG (difference in change, *p* = 0.04). There was no significant change in the consumption of vegetables	Some Concerns
Ickovics, 2019 ** [21]	___	Food environment and nutritional education (Pupils and parents)	3 years	Healthy foods (fruit, vegetables, green salad, potatoes—not fried) and unhealthy foods (French fries, chips, candy, ice cream, other sweets) (serving/day)	24 h dietary recall	Students (eighth grade) at schools randomized to the nutrition condition consumed fewer unhealthy foods (*p* < 0.03)	Low
Lytle, 2004 [22]	SCT	Classroom curriculum, school environment, and peer leaders. Parent Packs	2 years, 10 curriculum sessions per year	Fruits and vegetables (score)	24 h dietary recall	No significant differences	Some Concerns
Martens, 2008 [23]	___	Classroom and parents (a bag with information and food items for parents)	3 months, eight school classes lasting 50 min each	Fruit (serving/day)	Written questionnaires	The significant difference between groups at T1 = 0.04. *p* ≤ 0.05 two tailed	Some Concerns
Mihas, 2009 [24]	___	Pupils (classroom) and parents (seminars)	12 h of classroom material for 12 weeks	Fruits and vegetables (servings/day), ready-to-eat breakfast cereals, red meat, poultry, and fish (meals/week)	FFQ	A significant increase in poultry, ready-to-eat breakfast cereals, and fruit consumption and a significant decrease in red meat consumption were found in the IG. There was no significant difference in the consumption frequency of any food category in the CG	Some Concerns
Nicklas, 1998 [25]	___	(1) School-wide media-marketing campaign, (2) school-wide meal and snack modification, (3) classroom workshops and supplementary subject matter activities, and (4) parental involvement	3 years	FV (servings/day)	Knowledge, Attitudes, and Practices questionnaire **	No significant difference between groups	Low
Ochoa-Avilés, 2017 [26]	___	Matrices for adolescents, parents, and school staff	28 months. (30 min/day)	FV (servings/day) and unhealthy snacking (g/d)	24 h dietary recall	No significant difference between groups	Low
Rees, 2010 [29]	TPB and the Transtheoretical Model	Classroom	3 months	Fruits, vegetables, whole grains, brown bread (servings/day)	24 h dietary recall	The intervention group consumed approximately 0.35 more servings of brown bread weekly than the control group from baseline. Although this change between groups was statistically significant the magnitude was small. For the other foods, there were no significant effects of the tailored intervention	Some Concerns
Wang, 2015 [28]	Bronfenbrenner’s ecological theory	Holistic HP’s approach (school environment and ethos, modified curriculum, and family/community involvement)	3 months	Consumption score of fruit, vegetables, dairy products, breakfast, dessert, fried food, and soft drinks	FFQ	Students in the HPS school had the largest improvement in eating behaviors (from 3.16 to 4.13), followed by those in the HE school (from 2.78 to 3.54)	Some Concerns

* Food presented in the results of food consumption. ** Nutrition and physical activity interventions. FV—fruits and vegetables. SCT—Social Cognitive Theory. TPB—Theory of Planned Behaviour. HFSS—foods, and snack foods that are high in fats, sugar, and salt. 24 HR—24 h dietary recall daily.

## Data Availability

Not applicable.

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
