# Peer review of "Effect of School-Based Food and Nutrition Education Interventions on the Food Consumption of Adolescents: A Systematic Review and Meta-Analysis"

_ijerph, 2022, doi:10.3390/ijerph191710522_

Round 1

Reviewer 1 Report

The topic of this paper -- the effectiveness of school intervention programs to improve adolescents’ nutritional intake -- is an important one.

Major comments:

·       I recognise how difficult it is to write a paper in a language different from one’s first language; however, there are many places in this paper where the writing is difficult to follow.  I strongly recommend that a native English speaker edit the manuscript prior to re-submission.

·       The authors need to make a stronger argument at the start for the focus of this systematic review and meta-analysis.  They can’t simply state that something is important and then move on.  They state, for example, that “It is important that food and nutrition education interventions designed specifically for adolescents are evaluated, and that the review is more comprehensive in terms of both publication time and population, including studies from different countries with different socioeconomic conditions.”  But why?  Why is it important that food and nutrition education interventions designed specifically for adolescents be evaluated?  What is it about adolescence that requires such targeted interventions? The evidence for this needs to be discussed.  And why does the review need to be more comprehensive?  And more comprehensive than what? 

·       And again here: “The scanning [sic] was focused on the following four key elements: population (children, adolescents), intervention (Food and Nutrition Education interventions school-based), outcome (food consumption) and study design (clinical trial).”  Why these choices?  Also, I thought the focus was on adolescents, not children(?)

·       My other major concern is that the Discussion doesn’t really seem to match the focus of the paper/the results.  In the Discussion, the authors talk about the importance of multi-component interventions, knowledge about food and nutrition, theoretical models, and duration of interventions.  Yet, the Results section doesn’t really discuss any of this at all; it focuses on consumption of fruit and/or vegetables. Lines 414-418, for example, state: “it emphasizes out [sic] that interventions must be multicomponent, including changes in the environment and the participation of parents, teachers and school staff. Another important aspect is that the intervention should use theoretical model about behavioral changes, in addition to knowledge about food and nutrition.” Where is the support in the Results section for these statements?

Minor comments:

·       Why is each sentence a separate paragraph?  This makes the paper very difficult to read.  Paragraphs are intended to signal to the reader where there is a shift in the discussion.

·       In addition to language issues, there are a number of typo’s.  For example, on page 2, line 45, “not only” must be followed by “but also”; Lines 107-110: numerals under 10 should be written out;  Etc.

·       Lines 70-71: The authors just stated the failings of existing reviews, but then don’t state here how this review rectifies these deficiencies.

·       Lines 72-75: Are these the findings from this paper?  This was unclear to me.

·       Lines 87-89: I would state this at the start of the Methods section.  This was the approach used.

·       Lines 155-156: were all references checked?  I didn’t understand how the additional 14 references were identified.

·       Line 183: Why did the search start with 1997?  Or was this simply the first year a relevant paper was found?

·       Lines 205-206: Why is this bolded?

·       Line 230: What was the control intervention?

·       Line 254: It would be helpful to remind the reader how many of these studies there were

·       Line 260: Pre-post difference?

·       Line 260-262: I didn’t understand  “instead of seeking to increase knowledge”.  Isn’t this section about food consumption.

·       Line 286: What is meant by “inserted studies”?

·       Line 288: “As the consumption of fruits and vegetables is a marker of healthy food consumption” – this statement needs a reference

·       Lines 300-301: What are these two steps?

·       Lines 330-332: This sentence should have been in the intro

Reviewer 2 Report

Comments to the Authors

Effect of school-based food and nutrition education interventions on the food consumption of adolescents: a systematic review and meta-analysis

Text:

While the authors did a good job at justifying why their research question is worthwhile, I have some comments that might help this paper be even stronger:

1-    The effect on fruits was declared as neutral by the authors. I would phrase this differently saying the results were favorable for fruits and vegetable consumption increase but with more effect on vegetable consumption. The evidence on fruit consumption might be weaker due to the paucity of studies focusing on fruit consumption.

2-    The authors tend to write separate sentences separated in several paragraphs.  The authors should make an effort to make distinct paragraphs without to many sub-paragraphs. There needs to be a connection between the points and starting on a new line each time is not necessary.

3-    Risk of bias assessment section is missing in the methods.  In this section, authors should talk about study quality and why they were not able to assess small study bias.

4-    Study strengths is missing in the discussion.

Overall, this is a great paper and it will hopefully encourage policy makers to have more nutrition education interventions in schools.

Round 2

Reviewer 1 Report

I appreciate the authors' efforts to address my previous comments.  The resulting revised paper is now, in my opinion, much stronger in terms of its flow, focus, and comprehension.  The English is still a bit confusing in places -- particularly in added paragraphs -- but it is improved.  There are also still a number of typo's, but I will assume that the editorial staff will correct most of these.  My only remaining question is about the issue of multi-component versus single-component interventions.  Starting at line 60, the authors state that a prior systematic review concluded that multi-component interventions are more effective than single-component interventions.  I then expected some kind of a "Therefore" statement, such as "Therefore, our expectation was that our review would find a similar result."  Also, and it may be that I missed it, but did you indeed find that the multi-component interventions were more effective than the single-component interventions?

Author Response

Excellent question. We thank you for your careful reading and that has contributed a lot to the report of our study. Regarding your questioning, although we identified the characteristics of the interventions that had a positive effect on healthy food consumption, our analyzes do not allow us to say that the multi-component intervention is more effective than the single-component intervention. What we found is that multi-component interventions are more frequent among the interventions analyzed and have shown favorable effects on healthy food consumption. Multi-component interventions were more frequent than single-component interventions, probably because other studies have already shown the effectiveness of this type of intervention (Van et al., 2010; Mercado et al., 2016; Murimi et al., 2017; Murimi et al., 2017; Murimi et al. 2018).

Therefore, we have revised the text of the discussion and conclusion to avoid misinterpretation (lines 437-441; 470-471)

References

Van Cauwenberghe E, Maes L, Spittaels H, et al. Effectiveness of school-based interventions in Europe to promote healthy nutrition in children and adolescents: systematic review of published and “grey” literature. Br J Nutr. 2010;103:781–797

Mercado J, Aufa’l ARAM, Belyeu-Camacho T, et al. A review of promising multicomponent environmental child obesity prevention intervention strategies by the Children’s Healthy Living Program. J Environ Health. 2016;79:18–26.

Murimi MW, Kanyi M, Mupfudze T, et al. Factors influencing efficacy of nutrition education interventions: a systematic review. J Nutr Educ Behav. 2017;49:142–165

Murimi, M.W.; Moyeda-Carabaza, A.F.; Nguyen, B.; Saha, S.; Amin, R.; Njike, V. Factors that contribute to effective nutrition education interventions in children: A systematic review. Nutr Rev. 2018, Volume 76, pp. 553–80. DOI:10.1093/nutrient/nuy020